# Essential components of a definition for early antibiotic treatment failure: A scoping review

Hiroyoshi Iwata[1,2]*, Makoto Kaneko[3], Takuya Aoki[4], Koji Endo[5], Yuki Nagai[6], Kenji Kanto[7], Masahiro Yao[8], Shuhei Hamada[9,10]

1 Center for Environmental and Health Sciences, Hokkaido University, Sapporo, Hokkaido, Japan, 2 Center of Clinical Research and Quality Management, University of the Ryukyus Hospital, Nishihara, Okinawa, Japan, 3 Department of Health Data Science, Yokohama City University, Yokohama, Kanagawa, Japan, 4 Division of Clinical Epidemiology, Research Center for Medical Sciences, The Jikei University School of Medicine, Minatoku, Tokyo, Japan, 5 Department of Pharmacoepidemiology, Graduate School of Medicine and Public Health, Kyoto University, Kyoto, Japan, 6 Department of General Internal Medicine, National Hospital Organization Nagasaki Medical Center, Nagasaki, Japan, 7 Department of General Medicine, Ohara General Hospital, Fukushima, Japan, 8 Division of Family and Community Medicine, Yokohama Hodogaya Central Hospital, Japan Community Health Care Organization, Yokohama, Kanagawa, Japan, 9 Division of Clinical Medicine, Faculty of Medicine, University of Tsukuba, Tsukuba, Ibaraki, Japan, 10 Department of Internal Medicine, Kamisu Saiseikai Hospital, Kamisu, Ibaraki, Japan

* hiroyoshi-Iwata@cehs.hokudai.ac.jp

**Data Availability Statement:** Our protocol data are available from the OSF database: osf.io/d9wa4. DOI 10.17605/OSF.IO/D9WA4.

**Funding:** The authors received no specific funding for this work.

## Abstract

### Background

Despite the broad global use of antibiotics, there is no established definition of early antibiotic treatment failure (EATF) to aid clinical evaluation of treatment, which leads to inconsistent assessments of drug effectiveness.

### Aim

This scoping review aims to identify common components of EATF definitions by synthesizing studies mentioning EATF and its relevant thesaurus matches.

### Design

Scoping review

### Methods

This scoping review was conducted following the PRISMA Scoping review guidelines. A systematic literature search was conducted using MEDLINE (PubMed), CENTRAL, CINAHL, and Web of Science, as well as a manual Google search. Search terms were EATF and its thesaurus matches. After removing duplications, candidate studies were screened by title and abstract prior to full text searches, and quality analysis was performed on eligible studies using the Critical Appraisal Skills Programme. From each eligible study, the timing of evaluation, basic components, and detailed information for each definition of EATF were collected. The components of each definition for EATF were then summarized and counted, and finally the most common essential components were identified.

**Competing interests:** The authors have declared that no competing interests exist.

## Results

Our systematic literature search found 2,472 candidate studies. After title and abstract screening, full text search and quality assessment, 61 studies, including 56 original studies and five reviews, were eligible for our analysis. Of these 56 original studies, 43 mentioned the timing of EATF evaluation 72 hours after the start of treatment with antibiotics. From these 43 studies, the most common indicators of EATF were extracted, among which a set of essential components for a definition of EATF were identified: mortality, vital signs, fever, symptoms, and additional treatment.

## Conclusions

Our scoping review uncovered five essential factors for EATF. Further study is needed to evaluate the validity of our findings.

## Introduction

Antibiotics are administered with high frequency worldwide [1], but there is no established clinical standard definition for early antibiotic treatment failure (EATF) with criteria for assessing effectiveness in the early treatment phase [2, 3]. In most cases of infectious disease treatment, decisions about whether to continue, change, or discontinue antibiotics are made based on the individual clinician's personal criteria and discretion. Moreover, there are many cases in which it is difficult for clinicians to determine the clinical effectiveness of antibiotics. A finding of EATF can help prevent long-term use of unnecessary antibiotics, thereby reducing the risk of developing multidrug-resistant bacteria. In clinical research, EATF may also be warranted as an outcome of antibiotic treatment in observational or randomized controlled studies.

It is difficult to evaluate whether additional treatment succeeded or failed based on mortality alone. Some other outcomes of antibiotic treatment, such as hospital duration, are not applicable for early-stage evaluation. In the early stage of bacterial infectious disease, patients can experience critical events, including: shock, requiring vasopressors, and intubation for mechanical ventilation, in addition to death. Thus, a standardized set of criteria for evaluating early-stage antibiotic treatment effectiveness would have clinical utility.

There are several reports mentioning EATF. In 2009, Sánchez García published a review of "Early antibiotic treatment failure" [2]. However, that study did not conduct a systematic literature search, and over ten years has passed since it was published. There are other novel articles which mention "early antibiotic treatment failure" or its thesaurus matches, as described below. Bassetti *et al*. (2020) conducted a systematic review focused on the impact of appropriate versus inappropriate initial antibiotics therapy (IAT) [4]. Although EATF and IAT have some overlap, the IAT literature does not always refer to early-stage evaluation. To the best of our knowledge, the definitions of both EATF and IAT lack standardization and are still evolving. Recently, Rac *et al*. (2020) proposed a set of "early clinical failure criteria" composed of systolic blood pressure, heart rate, respiratory rate, altered mental status and white blood cell count [5]. However, this study investigated the criteria as a predictive variable of 28-day mortality, and did not present the evidence on which the criteria were based or an assessment of their validity.

Therefore, a scoping review was conducted in order to clarify the key components of EATF definitions based on the results of a systematic search of literature mentioning EATF and its thesaurus matches [6].

## Methods

### Search strategy and types of sources

Our protocol was registered with the Open Science Framework (OSF) on February 28[th], 2022 [7]. This study followed the Preferred Reporting Items for Systematic Reviews and Meta-analyses Extension for Scoping Reviews (PRISMA-ScR) and JBI Manual for Evidence Synthesis [8, 9]. (S1 Table) Our search covered all peer-reviewed publications published from January 1[st], 1980 to February 28[th], 2022. This is because the articles mentioning EATF in the Sánchez García EATF review article were from early 21[st] Century [2]. The search used Medline (PubMed), CENTRAL, CINAHL, Web of Science, and also performed a manual Google search. Our search terms are shown in S2 Table. Whenever possible, MeSH terms and keywords were used to increase the chances of finding relevant studies. The search strategy, including all identified keywords and index terms, was adapted for each included database. A librarian at the Jikei University School of Medicine supported the systematic literature search. The manual Google search also included literature cited in the eligible articles. Searches were not limited by language.

### Eligible publications

Any type of publication was included if it presented a conceptual definition of EATF or its thesaurus matches. This scoping review considered various study designs including randomized controlled trials, non-randomized controlled trials, before and after studies, interrupted time-series studies, prospective and retrospective cohort studies, case-control studies, cross-sectional studies, and case series studies. In addition, systematic reviews that meet the inclusion criteria were considered. Opinion and text papers, as well as research letters were also considered for inclusion. In contrast, case reports and animal studies were excluded.

### Participants/Study outcome

Our scoping review conducted a search targeting participants or studies whose outcome was EATF or its thesaurus matches.

### Inclusion and exclusion criteria

Studies presenting clear definitions for EATF or its thesaurus matches were included if evaluations were performed no later than seven days after admission or bacteremia onset or initiation of antibiotics treatment, among bacterial infection patients [2]. Because the definition of EATF is still evolving and there is overlap between EATF and "initial antibiotic treatment failure" (IATF), IATF studies were also included if their definition criteria mention outcomes no later than seven days after admission or bacteremia onset or initiation of antibiotics treatment. Finally, treatment success and its thesaurus matches, such as clinical response, were also included as a proxy for treatment failure.

Our exclusion criteria were: 1) studies whose target disease was not a bacterial infection, 2) studies whose target disease's standard treatment is not antibiotics, 3) studies whose target disease is resistant to common antibiotics, leading to a heightened risk of recurrence, such as *Mycobacterium* including tuberculosis, *Helicobacter*, and *Clostridium difficile*, 4) studies whose participants were pediatric patients or outpatients, 5) studies covering only oral treatment, 6) studies not reporting a clear definition of EATF or IATF, and 7) studies not reporting the timing of their evaluations, or whose evaluations were conducted more than seven days after admission or bacteremia onset or initiation of antibiotics treatment.

## Literature search, data extraction process and quality assessment

After combining literature search results and removing duplications, two independent teams [HI] and [KE, SY] performed the first screening using titles and abstracts. Next, HI, KE and MK performed full text reviews. Discrepancies of inclusion or exclusion were resolved by all authors. After choosing candidates with two independent teams, we [YN, KK, SH] independently performed quality analysis using the Critical Appraisals Skill Programme (CASP) [10, 11]. TA audited the whole process from an objective viewpoint. EndNote X9.3.3 was used to sort the literature and remove duplications.

## Data presentation and identification of common components of EATF definitions

After the literature search and quality assessment, the definitions of EATF were compiled in a table. The most frequently used timing of EATF assessment was identified from among the eligible studies, excluding reviews. Next, because different timing of evaluation is likely to result in different events constituting failure, the investigation was narrowed to studies reporting the most frequent timing of evaluation. Among the eligible studies, excluding reviews, using the most frequently reported timing for EATF assessment, common components which comprise EATF definitions were identified. Finally, the common essential components of a definition for EATF were summarized.

The components were divided into several categories: mortality, symptoms, vital signs, body temperature, disease-specific changes, radiographic changes, persistent positive blood culture, need of additional treatment including new antibiotics and intubation for mechanical ventilation and vasopressors, new onset of other disease, source control (drainage or operation), intensive care unit (ICU) admission, readmission, and recurrence of the original disease. While body temperature (fever) is itself a vital sign, high fever alone can be an indispensable clinical factor [12]. Hence, vital signs and body temperature are listed separately. Common elements of the definitions for EATF found in eligible studies which measured EATF at the most frequent timing were then summarized.

## Results

Our study flow chart is shown in Fig 1. Our systematic literature search produced 2472 potentially eligible studies. Sixteen articles were added through a manual search on Google and Google Scholar. After removing duplications, 2,088 studies were screened by title and abstract, after which 168 studies were deemed appropriate for the full text assessment. In the full text assessment, 107 studies were found to be ineligible for the following reasons: no EATF definition (46), no definition of early phase outcome (48), not bacterial disease (6), pediatrics study (3), case report (2), unavailable in domestic library network (2). After the full-text assessment, 56 studies were eligible for the qualitative assessment using the CASP to assess the risk of bias and applicability, in addition to which five review articles were included. Finally, 61 studies, including the five reviews, were eligible for our analysis. The references of the eligible studies are shown in S3 Table. No authors were contacted.

The characteristics of the eligible studies (ID 1–61) and their EATF components are summarized in Table 1. The first mention of EATF was found in a study published in 1994. All included studies were written in English, except Wang *et al.* (2014) [13], which was written in Chinese. The most common countries of publication were Sweden, South Korea, and the United States. The most common research target disease was pneumonia (including community-acquired pneumonia and ventilator-associated pneumonia), followed by urinary tract infection. Common EATF thesaurus matches included clinical success, clinical response,

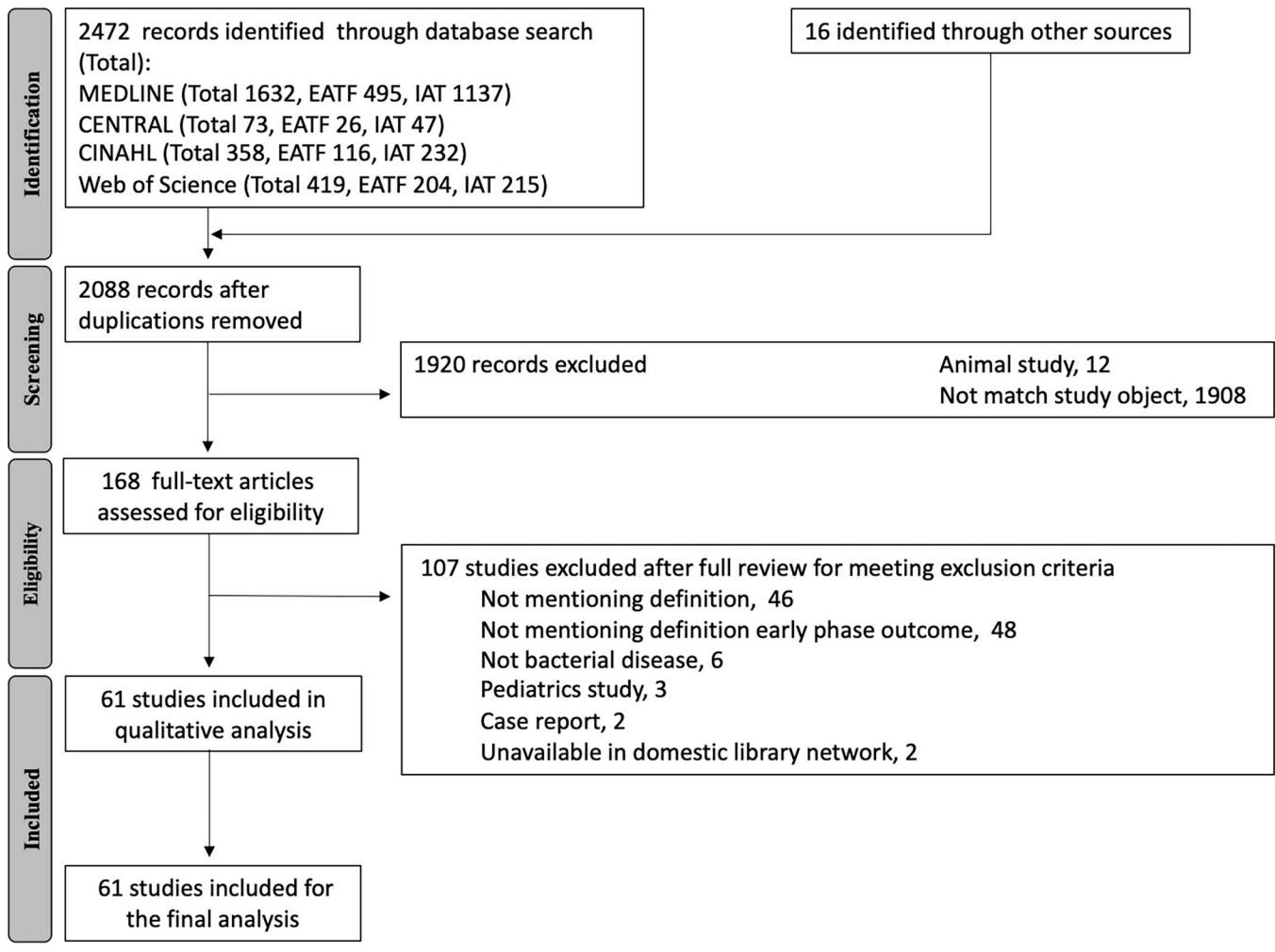

**Fig 1. PRISMA literature search flowchart.**

clinical failure, initial treatment failure, and early treatment failure. Eight studies reporting infections which may need additional source control were identified, including complicated skin and skin-structure infection, complicated urinary tract infection, acute appendicitis, intra-abdominal infection, and diverticular abscess. However, because these infections can be successfully treated with antibiotics, these eight studies were not excluded from our analysis.

Eligible studies used varying definitions for EATF and evaluations were performed at various times. Aside from the five review studies, of the 56 other eligible studies, 43 mentioned EATF evaluation at 72 hours after the initiation of antibiotics treatment, suggesting that the most common timing for evaluating EATF is 72 hours.

Next, the components of the EATF definitions were extracted. (Table 2) Major components of definitions for EATF were found to be mortality, symptoms, vital signs, fever, and need of additional treatment.

## Discussion

The present study is a systematic scoping review, covering 56 original studies and five reviews mentioning EATF. The most frequent timing for EATF evaluation was 72 hours after the

**Table 1. Characteristics of eligible studies and components of definitions for early antibiotics treatment failure.**

| Study ID | Author | Outcome name | Year | Language | Country | Disease | 48 hours | 72 hours | Day 4 | Day 5 | Day 7 | Type |
|---|---|---|---|---|---|---|---|---|---|---|---|---|
| 1 | Erjavec *et al.* | Response rate | 1994 | English | Netherlands | Various | | ○ | | | | original |
| 2 | Bosi *et al.* | Success, failure | 1999 | English | Italy | Various | | ○ | | | | original |
| 3 | Arancibia *et al.* | Failure | 2000 | English | Spain | Pneumonia (community-acquired pneumonia) | | ○ | | | | original |
| 4 | Ioanas *et al.* | Nonresponse | 2004 | English | Spain | Pneumonia (intensive care unit–acquired pneumonia) | | ○ | | | | original |
| 5 | Menendez *et al.* | Early treatment failure | 2004 | English | Spain | Pneumonia (community-acquired pneumonia) | | ○ | | | | original |
| 6 | Edelsberg *et al.* | Treatment failure | 2008 | English | United States | Complicated skin and skin-structure infections* | | ○ | | | ○ | original |
| 7 | Bruns *et al.* | Early clinical failure | 2009 | English | Netherlands | Pneumonia (community-acquired pneumonia) | | ○ | | | | original |
| 8 | Mitja *et al.* | Early mortality | 2009 | English | Spain | Listeriosis | | ○ | | ○ | | original |
| 9 | Shindo *et al.* | Initial treatment failure | 2009 | English | Japan | Pneumonia (health-care-associated pneumonia) | ○ | ○ | | | ○ | original |
| 10 | Cheng *et al.* | Clinical response | 2010 | English | Taiwan | Gram negative bacterial infections | | ○ | | | | original |
| 11 | Tumbarello *et al.* | Treatment failure, initial response to treatment | 2010 | English | Italy | Bacteremia (bloodstream infections) | | ○ | | | | original |
| 12 | Vogelaers *et al.* | Clinical response | 2010 | English | Germany | Severe nosocomial infections | | | | ○ | | original |
| 13 | Yakar *et al.* | Antibiotic failure | 2010 | English | Turkey | Spontaneous bacterial peritonitis | | ○ | | | | original |
| 14 | Jeon *et al.* | Initial treatment failure | 2011 | English | Korea | Pneumonia | ○ | ○ | | | ○ | original |
| 15 | Stojadinovic *et al.* | Early clinical failure | 2011 | English | Serbia | Kidney Infections | | | ○ | | | original |
| 16 | Waltner-Toews *et al.* | Early clinical response | 2011 | English | United States | Bacteremia (bloodstream infections) | ○ | ○ | ○ | ○ | | original |
| 17 | Eckburg *et al.* | Clinical response | 2012 | English | Various (Africa, Asia, Eastern Europe, Western Europe, Latin America, and the United States) | Pneumonia (community-acquired pneumonia) | | ○ | | | | original |
| 18 | Janisch *et al.* | Failure | 2012 | English | Germany | Various | | ○ | | | | original |
| 19 | O'Neal *et al.* | Treatment failure | 2012 | English | United States | Bacteremia | | ○ | | | | original |
| 20 | Ott *et al.* | Treatment failure | 2012 | English | Germany | Pneumonia | | ○ | | | | original |
| 21 | Berger *et al.* | Initial treatment failure | 2013 | English | United States | Complicated skin and skin-structure infections* | | ○ | | | | original |
| 22 | Kang *et al.* | Unfavorable treatment response | 2013 | English | Korea | Bacteremia | ○ | | | | | original |
| 23 | Maruyama *et al.* | Initial treatment failure | 2013 | English | Japan | Pneumonia | ○ | ○ | ○ | ○ | ○ | original |

(*Continued*)

**Table 1.** (*Continued*)

| Study ID | Author | Outcome name | Year | Language | Country | Disease | 48 hours | 72 hours | Day 4 | Day 5 | Day 7 | Type |
|---|---|---|---|---|---|---|---|---|---|---|---|---|
| 24 | Robinson *et al.* | Treatment response | 2014 | English | United States | Pneumonia (community-acquired pneumonia) | | | ○ | | | original |
| 25 | Saverio *et al.* | Short-term efficacy of antibiotic treatment, failure | 2014 | English | Italy | Acute Appendicitis* | | | | | ○ | original |
| 26 | Wang *et al.* | Early treatment failure | 2014 | Chinese | China | Pneumonia (community-acquired pneumonia) | | ○ | | | | original |
| 27 | Wie *et al.* | Early clinical success, early clinical failure | 2014 | English | South Korea | Urinary tract infection (Pyelonephritis) | | ○ | | | | original |
| 28 | Wie *et al.* | Early clinical success, early clinical failure | 2014 | English | South Korea | Urinary tract infection (Pyelonephritis) | | ○ | | | | original |
| 29 | Chong *et al.* | Failure of initial antibiotic therapy | 2015 | English | South Korea | Intra-abdominal infections | | | | | ○ | original |
| 30 | Elagili *et al.* | Treatment failure | 2015 | English | United States | Diverticular abscess* | ○ | | | | | original |
| 31 | Lodise *et al.* | Clinical response | 2015 | English | United States | Pneumonia (community-acquired pneumonia) | ○ | ○ | ○ | ○ | ○ | original |
| 32 | Torres *et al.* | Early treatment failure | 2015 | English | Spain | Pneumonia (community-acquired pneumonia) | | ○ | | | | original |
| 33 | Hsieh *et al.* | Early clinical failure | 2016 | English | Taiwan | Bacteremia | | ○ | | | | original |
| 34 | Jääskeläinen *et al.* | Treatment failure | 2016 | English | Finland, Sweden | Complicated skin and skin-structure infections* | | | | ○ | | original |
| 35 | Merli *et al.* | Treatment failure | 2016 | English | Italy | Health-care-associated infections among cirrhosis patients | | ○ | | | | original |
| 36 | Park *et al.* | Clinical success, clinical failure | 2016 | English | South Korea | Urinary tract infection (Pyelonephritis) | | ○ | | | | original |
| 37 | Ramirez *et al.* | Clinical response | 2016 | English | Spain | Pneumonia (ventilator-associated pneumonia) | | ○ | | | | original |
| 38 | Babich *et al.* | Clinical failure | 2017 | English | Israel | Urinary tract infection (Catheter-Associated) | | | | | ○ | original |
| 39 | Ceccato *et al.* | Early treatment failure | 2017 | English | Spain | Pneumonia (community-acquired pneumonia) | ○ | ○ | | | | original |
| 40 | Ereshefsky *et al.* | Clinical cure | 2017 | English | United States | Serious nosocomial infections | | | | | ○ | original |
| 41 | Lee *et al.* | Clinical response | 2017 | English | Taiwan | Bacteremia | | ○ | | | | original |
| 42 | Ruiz-Ramos *et al.* | Treatment failure, clinical response | 2017 | English | Spain | Pneumonia (ventilator-associated pneumonia) | | ○ | | | | original |
| 43 | Trupka *et al.* | Early failure | 2017 | English | United States | Pneumonia (ventilator-associated pneumonia) | ○ | | | | | original |
| 44 | El-Sokkary *et al.* | Clinical response | 2018 | English | Egypt | Pneumonia (community-acquired pneumonia) | ○ | ○ | | | | original |

(*Continued*)

**Table 1.** (Continued)

| Study ID | Author | Outcome name | Year | Language | Country | Disease | 48 hours | 72 hours | Day 4 | Day 5 | Day 7 | Type |
|---|---|---|---|---|---|---|---|---|---|---|---|---|
| 45 | Karve *et al.* | Success, failure, intermediate | 2018 | English | Brazil, France, Italy, Russia, Spain | Urinary tract infections (complicated urinary tract infection) * | ○ | | | | | original |
| 46 | Nie *et al.* | Treatment failure | 2018 | English | China | Pneumonia (community-acquired pneumonia) | | ○ | ○ | ○ | ○ | original |
| 47 | Eliakim-Raz *et al.* | Treatment failure | 2019 | English | 20 countries in Europe and the Middle East | Urinary tract infection (complicated urinary tract infection) * | | | | ○ | ○ | original |
| 48 | Kim SH *et al.* | Treatment failure, clinical treatment success | 2019 | English | South Korea | Urinary tract infection (Pyelonephritis) | | | | | ○ | original |
| 49 | Peeters *et al.* | Initial treatment failure, treatment success, treatment failure | 2019 | English | Brazil, France, Italy, Russia, Spain | Intra-abdominal infection* | ○ | | | | | original |
| 50 | Wongsurakiat *et al.* | Early treatment failure | 2019 | English | Thailand | Pneumonia (community-acquired pneumonia) | ○ | ○ | | | | original |
| 51 | Al-Hasan *et al.* | Early treatment failure | 2020 | English | United States | Bacteremia (gram-negative bloodstream infections) | | ○ | | | | original |
| 52 | Kim YJ *et al.* | Early clinical response | 2020 | English | South Korea | Urinary tract infection | | ○ | | | | original |
| 53 | Rac *et al.* | Early clinical failure | 2020 | English | United States | Bacteremia (gram-negative bloodstream infections) | | ○ | ○ | | | original |
| 54 | Shimoni *et al.* | Response to antibiotic therapy | 2020 | English | Israel | Urinary tract infection | | ○ | | | | original |
| 55 | Herrmann *et al.* | Early treatment response | 2021 | English | Germany | Bacteremia (Bloodstream Infections) | | ○ | | | | original |
| 56 | Mun *et al.* | Early antibiotic treatment failure | 2021 | English | South Korea | Bacteremia | | ○ | | | | original |
| 57 | Garcia-Vidal *et al.* | Early failure | 2009 | English | Review article | Pneumonia (community-acquired pneumonia) | | ○ | | | | Review article |
| 58 | Sánchez García M | Early antibiotic treatment failure | 2009 | English | Review article | Various | | | | | | Review article |
| 59 | Cao *et al.* | Failed initial therapy | 2018 | English | Review article | Pneumonia (community-acquired pneumonia) | | ○ | | | | Review article |
| 60 | Bassetti *et al.* | Treatment failure | 2020 | English | Review article | Various (severe bacterial infections) | | | | | | Review article |
| 61 | Ceccato *et al.* | Clinical response | 2022 | English | Review article | Pneumonia (ventilator-associated pneumonia) | | ○ | | | | Review article |

○; Applicable

* Target disease may need infection control.

**Table 2. Components of definitions for early antibiotics treatment failure evaluated 72 hours after antibiotics treatment.**

| Study ID | Author | Outcome | Mortality | Symptom | Vital sign | Mental status | Fever | Laboratory change | Radiographic change | Need additional therapy | Disease specific change | Source control (Drainage or operation) | New disease onset | ICU admission | Multiple organ failure | Persistent blood culture | Readmission | Recurrence |
|---|---|---|---|---|---|---|---|---|---|---|---|---|---|---|---|---|---|---|
| 1 | Erjavec et al. | Response rate | | ○ | ○ | | ○ | ○ | | | | | | | | | | |
| 2 | Bosi et al. | Success, failure | | | | | ○ | | | ○ | | | | | | | | |
| 3 | Arancibia et al. | Failure | | ○ | | | ○ | | | | | | | | | | | |
| 4 | Menendez et al. | Early treatment failure | | | ○ | | ○ | | ○ | ○ | | | ○ | | | | | |
| 5 | Ioanas et al. | Non response | | | ○ | | ○ | | ○ | ○ | | | | | | ○ | | |
| 6 | Edelsberg et al. | Treatment failure | | | | | | | | ○ | | ○ | | | | | | |
| 7 | Mitja et al. | Early mortality | ○ | | | | | | | | | | | | | | | |
| 8 | Bruns et al. | Early clinical failure | ○ | | ○ | ○ | ○ | | | ○ | | | | ○ | | | | |
| 9 | Shindo et al. | Initial treatment failure | | ○ | ○ | | ○ | | ○ | ○ | | | | | | | | |
| 10 | Cheng et al. | Clinical response | | ○ | ○ | | | | | ○ | | | | | | | | |
| 11 | Tumbarello et al. | Treatment failure, initial response to treatment | ○ | ○ | ○ | | | | | | | | | | | | | |
| 13 | Yakar et al. | Antibiotic failure | | ○ | ○ | | | | | | ○ | | | | | | | |
| 14 | Jeon et al. | Initial treatment failure | ○ | ○ | ○ | | ○ | | | ○ | | | | | | | | |
| 16 | Waltner-Toews et al. | Early clinical response | | | | | ○ | ○ | | ○ | | | | | | | | |
| 17 | Eckburg et al. | Clinical response | | ○ | ○ | ○ | ○ | | | | | | | | | | | |
| 18 | Janisch et al. | Failure | | | | | | ○ | | | | | | | | | | |
| 19 | O'Neal et al. | Treatment failure | | | | | ○ | ○ | | ○ | | | | | | ○ | | |
| 20 | Ott et al. | Treatment failure | | | | | | | | ○ | | | | | | | | |
| 21 | Berger et al. | Initial treatment failure | | | | | | | | ○ | | ○ | | | | | | |
| 23 | Maruyama et al. | Initial treatment failure | | | | | ○ | | ○ | ○ | ○ | | | | | | | |
| 26 | Wang et al. | Early treatment failure | ○ | ○ | ○ | | ○ | | | ○ | | | | | | | | |
| 27 | Wie et al. | Early clinical success, early clinical failure | | ○ | ○ | | ○ | | | | ○ | | | | | | | |
| 28 | Wie et al. | Early clinical success, early clinical failure | | ○ | ○ | | ○ | | ○ | | ○ | | | | | | | |
| 31 | Lodise et al. | Clinical response | ○ | ○ | | ○ | | | | | | | | | | | | |

*(Continued)*

**Table 2.** (Continued)

| Study ID | Author | Outcome | Mortality | Symptom | Vital sign | Mental status | Fever | Laboratory change | Radiographic change | Need additional therapy | Disease specific change | Source control (Drainage or operation) | New disease onset | ICU admission | Multiple organ failure | Persistent blood culture | Readmission | Recurrence |
|---|---|---|---|---|---|---|---|---|---|---|---|---|---|---|---|---|---|---|
| 32 | Torres et al. | Early treatment failure | ○ | | ○ | | | | | ○ | | | | | | | | |
| 33 | Hsieh et al. | Early clinical response | ○ | | | | | | | ○ | | | | | | | | |
| 35 | Merli et al. | Treatment failure | | | | | | ○ | | | ○ | | | | | ○ | | |
| 36 | Park et al. | Clinical success, clinical failure | | ○ | | | ○ | | | | | | | | | | | |
| 37 | Ramirez et al. | Clinical response | | | ○ | | ○ | | ○ | | ○ | | | | | | | |
| 39 | Ceccato et al. | Early treatment failure | ○ | | ○ | | | | | ○ | | | | | | | | |
| 41 | Lee et al. | Clinical response | ○ | | ○ | | | | | ○ | | | | | | | | |
| 42 | Ruiz-Ramos et al. | Treatment failure, clinical response | | ○ | ○ | | ○ | ○ | ○ | ○ | ○ | | | | | | | |
| 44 | El-Sokkary et al. | Clinical response | ○ | | ○ | | ○ | | ○ | | | | | | | | | |
| 46 | Nie et al. | Treatment failure | ○ | | | | | | | ○ | | | | ○ | | | | |
| 50 | Wongsurakiat et al. | Early treatment failure | ○ | | ○ | | | | | ○ | | | | | | | | |
| 51 | Al-Hasan et al. | Early treatment failure | ○ | ○ | ○ | ○ | ○ | ○ | | ○ | | | | | | | | |
| 52 | Kim YJ et al. | Early clinical response | | ○ | ○ | | ○ | ○ | | | | | | | | | | |
| 53 | Rac et al. | Early clinical failure | | | ○ | ○ | | ○ | | | | | | | | | | |
| 54 | Shimoni et al. | Response to antibiotic therapy | | | | | | | | | | | | | | | | |
| 55 | Herrmann et al. | Early treatment response | ○ | | ○ | | | ○ | | | ○ | | | | | | | |
| 56 | Mun et al. | Early antibiotic treatment failure | ○ | | ○ | | ○ | | | ○ | | | | | | | | |

○; Applicable, ICU; Intensive care unit

initiation of antibiotics treatment. Our scoping review identified the five most common EATF components: mortality, vital signs, fever, symptoms, and additional treatment.

EATF is a potentially useful framework for clinicians and researchers in various fields. In clinical practice, EATF can be used to judge the effectiveness of antibiotics. Effective EATF evaluation may help prevent unnecessary changes to broad-spectrum antibiotics, lower the risk of Clostridium difficile colitis, prevent the development of multidrug-resistant bacteria, and improve mortality [14, 15]. EATF can also allow researchers to assess the effectiveness of target antibiotics in clinical research. EATF may also have applications in the assessment of outcomes in antimicrobial drug trials and the establishment of criteria for early discontinuation. Furthermore, the appropriateness of the EATF components depends on the application. For example, the mortality component would be of no value in informing clinical treatment decisions. Moreover, clinical practice may benefit from similar definitions for treatment failure in fungal infections, tuberculosis, and viral diseases such as COVID-19. Further research is needed to establish definitions for early-stage assessment of these conditions. Furthermore, as the concept of EATF continues to be refined, methods for reaching consensus on a definition should be also considered. One possibility is the Delphi method, which facilitates consensus-building and minimizes the influences of potential sources of bias such as conflicts of interest and interpersonal relationships [16].

The common components for defining EATF identified in this study comport with definitions presented in previous reports. Among the five common components, vital signs were the most common components, followed by additional treatment, mortality, fever and symptoms (Table 2). To the authors' knowledge, the earliest EATF review article was published by Sánchez García (2009) [2]. That article compiled criteria used to diagnose treatment failure in a table. The components identified all match the criteria which he presented. Mun *et al.* (2021) included EATF in their title and presented a definition for EATF [17]. Their criteria for EATF comprise mortality, vital signs, fever, and additional treatment, which are consistent with our findings.

Among the five common components identified, vital signs, fever, and mortality were measured objectively, while there was some variation in additional treatments and much variation in symptoms. Additional treatments included the use of vasopressors and mechanical ventilation, which are standard advanced medical care for many diseases. However, symptoms depend on the original disease. Furthermore, because responses to antibiotic treatment may be influenced by resistance patterns in specific geographical areas, the results of antibiotics treatment studies may not be generalizable across populations with vastly different levels of resistance. Our systematic literature was conducted without limiting for geographical area, clinical setting, local antibiotic resistance pattern, or language. Future studies of EATF should adjust the definition of symptoms according to the research target and attempt to account for the influence of local antibiotic resistance patterns. Finally, because this study excluded target diseases whose standard treatment is not antibiotics, source control was not included among the factors for analyzing EATF which were identified. If the concept of EATF were expanded to cover all types of bacterial infections, source control would be an indispensable consideration from the perspective of avoiding antibiotic escalation.

Given the most common timing of EATF evaluation uncovered in our study, 72 hours appears to be a generally accepted timeframe. However, various timeframes are used for EATF evaluation among the studies included in our review, and there is no concrete evidence to support evaluation at 72 hours [2]. The Centers for Disease Control and Prevention provides "Core Elements of Hospital Antibiotic Stewardship Programs" which mention antibiotic time-outs at 48–72 hours of treatment to facilitate appropriate antibiotic selection [18, 19]. Rac *et al.* (2020) mentioned evaluations performed at 72–96 hours, which also largely matches our findings, except for the inclusion of altered mental status and white blood cell count [5].

The present study systematically surveyed existing definitions for EATF. However, EATF remains a nascent concept, and there are several ways in which it may be improved for future application. First, improved rapid diagnostic methods now enable clinicians to quickly find multidrug-resistant organisms in blood cultures. Among the eligible studies discussing bacteremia, there were no differences in definitions for EATF based on whether the involved organism is Gram-positive or Gram-negative. However, responses to antibiotics and clinical courses are often different for Gram-positive and Gram-negative bacteria. Thus, developing different definitions for EATF based on Gram-negative bacteria (especially multidrug-resistant strains) and Gram-positive bacteria may be clinically useful. Second, while bacterial count may influence clinical outcomes in some cases, none of the 61 eligible studies included bacterial count as a factor for EATF. However, Mun et al. (2021) suggests the possibility of bacterial count affecting clinical outcomes and the importance of source control with some organisms [17]. Future studies may evaluate the importance of bacterial count (inoculum effect) and its potential relevance for assessing EATF. Lastly, this study comprehensively collected a diverse set of definitions for EATF from various studies. While each of these has its advantages, such as often being tailored to the target disease, they also have distinct disadvantages, such as being difficult to define based on observation of symptoms or other objective measures, and being difficult to apply to diseases other than the target disease. Therefore, a precise universal definition that can be applied to all types of bacterial infections may be difficult to achieve. Accordingly, the present study attempts to identify the essential components of a definition for EATF, rather than prescribing a definitive formulation.

There are some limitations to our study. First, specific definitions for each component of EATF could not be identified; individual definitions were vague or inconsistent, and none of the five components converged on a unified definition. Further investigation is needed to define the details of each component and evaluate their validity. Next, the literature search was not conducted using EMBASE, due to financial constraints. In order to overcome this limitation, a range of common systematic literature search methods were employed. An experienced research librarian provided assistance with the selection of search terms and formulae, and two independent teams conducted manual searches on Google and Google Scholar. We also searched for all papers referenced in the reviews and articles that were uncovered with mentions of EATF. Further, the definition of EATF falls within the broader concept of treatment failure. For example, if treatment failure is defined as mortality within seven days after treatment initiation, this may also meet the definition of EATF. Therefore, some potentially eligible studies may have been left out of the present review. However, our scoping review aims to identify common components used to define EATF, rather than to synthesize numerical figures as in a meta-analysis. Moreover, there are many vague definitions, particularly for symptoms, among the 56 studies in the present review. However, symptoms should be included as a common component because it appears in more than 40% of the eligible studies. Further, many types of bacterial infections were included, which contributes to heterogeneity among the definitions of EATF. While there is much variety among diseases, the present study assessed common physical changes caused by bacterial infections in the hope that these findings would apply to a range of bacterial diseases. Finally, the protocol for this study was not registered on PROSPERO because scoping reviews are no longer eligible for registration as of as of February 2022 [20]. The protocol was therefore registered with the OSF, as suggested by JBI [9].

## Conclusions

Our scoping review identified the common components of EATF: mortality, vital signs, fever, symptoms, and additional therapy, evaluated 72 hours after the initiation of antibiotics

treatment. Further studies are needed to define the details of each component of EATF and investigate its validity.

## Supporting information

**S1 Table. PRISMA checklist.**
(DOCX)

**S2 Table. MEDLINE (PubMed), CENTRAL, CINAHL, Web of Science search terms and results.**
(DOCX)

**S3 Table. Reference list for eligible studies.**
(DOCX)

## Acknowledgments

The authors would like to thank Izumi Osaki at the Academic Information Center of the Jikei University School of Medicine for systematic literature search support, Keisuke Kamada at the Research Institute of Tuberculosis, Japan Anti-Tuberculosis Association's Department of Mycobacterium Reference and Research for expert advice related to infectious diseases and review and editing of the manuscript, Shinichiro Ueda at the University of Ryukyus for general research advice, and Allen Paul Heffel for writing assistance.

## Author Contributions

**Conceptualization:** Hiroyoshi Iwata, Makoto Kaneko, Takuya Aoki, Koji Endo, Yuki Nagai, Kenji Kanto.

**Data curation:** Hiroyoshi Iwata, Makoto Kaneko, Koji Endo, Yuki Nagai, Kenji Kanto, Masahiro Yao, Shuhei Hamada.

**Formal analysis:** Hiroyoshi Iwata, Makoto Kaneko, Takuya Aoki, Yuki Nagai, Kenji Kanto, Shuhei Hamada.

**Methodology:** Hiroyoshi Iwata, Makoto Kaneko, Takuya Aoki, Koji Endo, Yuki Nagai, Kenji Kanto.

**Resources:** Hiroyoshi Iwata, Takuya Aoki, Yuki Nagai, Kenji Kanto.

**Software:** Hiroyoshi Iwata.

**Supervision:** Makoto Kaneko, Takuya Aoki, Masahiro Yao.

**Validation:** Hiroyoshi Iwata, Makoto Kaneko, Shuhei Hamada.

**Writing – original draft:** Hiroyoshi Iwata, Makoto Kaneko, Takuya Aoki, Koji Endo, Kenji Kanto, Masahiro Yao, Shuhei Hamada.

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
