## [Decision Letter · Decision Letter 0]

10 Feb 2023

PONE-D-22-31074Essential components of a definition for early antibiotic treatment failure: a scoping review.PLOS ONE

Dear Dr. Hiroyoshi Iwata,

Thank you for submitting your manuscript to PLOS ONE. After careful consideration, we feel that it has merit but does not fully meet PLOS ONE’s publication criteria as it currently stands. Therefore, we invite you to submit a revised version of the manuscript that addresses the points raised during the review process.

We look forward to receiving your revised manuscript.

Kind regards,

Ali Amanati

Academic Editor

PLOS ONE

Journal Requirements:

"No"

Additional Editor Comments (if provided):

Dear authors

Your manuscript [ID Number PONE-D-22-31074] has passed through the review stage and is ‎ready for revision. ‎

Editorial comments

To ensure the Editor and Reviewers can recommend that your revised manuscript is ‎accepted, ‎please pay careful attention to each of the comments posted underneath ‎this email. This way we ‎can avoid future rounds of clarifications and revisions, moving swiftly to ‎a decision.‎

‎1. Please provide a point-by-point response to the the Editor and reviewer's comments

‎2. Please highlight all the amends on your manuscript with yellow colour

‎3. Minor English language correction is needed

‎

Other shortcomings

First of all, early antibiotics treatment response could be interpreted only ‎according to the local antibiotic resistance pattern and the geographical area of ‎the conducted studies. Therefore, the findings of studies obtained in areas with ‎low levels of resistance cannot be generalized for other areas with high levels of ‎resistance (lack of external validity). This was expected to be included in the ‎discussion.

Second, due to the improvement of rapid diagnostic methods in ‎bacterial infections, such as the use of rapid detection methods for multi-‎resistant bacteria in blood cultures, these definitions can be different based on ‎Gram-negative (especially multi-drug resistant Gram-negative) and Gram-‎positive organisms.

Tthird, bacterial infections are also divided based on the ‎bacterial inoculum size and therefore should be considered in the early ‎antibiotic’s treatment failure definition. While urinary tract infections have a ‎low bacterial inoculum size, pneumonia and sepsis have a high bacterial ‎inoculum size, and thus the response to treatment may be different. Overall, for ‎serious bacterial infections more stringent definitions should be considered; ‎however, for milder infections, more flexible definitions may be acceptable. ‎There is no discussion in this regard in the manuscript and it is recommended ‎to improve the discussion section accordingly. Also, if possible, a table can be ‎prepared (based on the definitions of bacterial inoculum size) for more precise ‎comparison.

Although the findings of this study could be helpful there are ‎many shortcomings in the definition, which will make it difficult to reach an ‎accurate definition based on the wide variety of bacterial infections‏.‏

Reviewers' comments:

Reviewer's Responses to Questions

**Comments to the Author**

1. Is the manuscript technically sound, and do the data support the conclusions?

Reviewer #1: Yes

Reviewer #2: Yes

2. Has the statistical analysis been performed appropriately and rigorously? 

Reviewer #1: N/A

Reviewer #2: Yes

3. Have the authors made all data underlying the findings in their manuscript fully available?

Reviewer #1: Yes

Reviewer #2: Yes

4. Is the manuscript presented in an intelligible fashion and written in standard English?

Reviewer #1: Yes

Reviewer #2: Yes

5. Review Comments to the Author

Reviewer #1: Congratulations on the paper. The subject is exciting, and a lot of work to make this review is apparent.

Significant revisions/clarifications:

- Elaborate further about the potential uses of EATF. However, this is not the purpose of the review; discussing the possible uses would be helpful for readers to integrate the concept into their work. Also, clarifying the potential uses even has implications for the components of the definition. For example, the use of EATF as an outcome for RCTs for new antibiotics has essential differences when compared with the use for clinical or antimicrobial stewardship purposes. For the second, component mortality would be of little help.

- From an antimicrobial stewardship point of view, using EATF raises some concerns that should be addressed. For example, analyzing together studies that report infections that need source control and those that typically do not need may create a concept (EATF) that promotes antibiotic escalation when the problem is source control. The authors should separate their findings for infections that typically do not need source control, such as respiratory tract infections and those that usually need source control.

- Elaborate further about future steps to develop this interesting EATF concept and how it can be studied and used in future. Would the authors suggest some consensus procedure to define the concept? Delphi, for example?

Minor revision:

- Page 21, line 223: mortality is misspelt.

Reviewer #2: The manuscript by Hiroyoshi Iwata et al. answers an important definitional question on "Early antibiotics treatment failure" by a systematic review of the literature.

This manuscript is well-written and describes a well-conducted systematic review but deserves rare and formative revisions before possible acceptance for publication.

Italicize "et al." and names of bacteria

Prefer passive forms.

Why didn't the authors use EMBASE? Numbers less than or equal to 12 should be spelled out if appropriate.

6. PLOS authors have the option to publish the peer review history of their article (what does this mean?). If published, this will include your full peer review and any attached files.

Reviewer #1: No

Reviewer #2: No

---

## [Author Response · Author response to Decision Letter 0]

3 Mar 2023

Dr. Ali Amanati

March 1, 2023

Dear Dr. Amanati,

On behalf of all of the authors, I would like to sincerely thank you and your team for your comments and suggestions for the improvement of our manuscript. We believe we have been able to apply them in order to constructively refine and strengthen our paper. We respond to the individual comments from both you and the reviewers below.

Regarding our funding, we apologize for the lack of clarity in the financial disclosure which accompanied our initial submission. We have updated our cover letter – sent separately – to include the statement: “The authors received no specific funding for this work,” which correctly characterizes our lack of funding for this study.

Finally, Dr. Keisuke Kamada provided us with some assistance in the revision of this manuscript. He has been added to our acknowledgements section accordingly.

Editor Comments

“First of all, early antibiotics treatment response could be interpreted only ‎according to the local antibiotic resistance pattern and the geographical area of ‎the conducted studies. Therefore, the findings of studies obtained in areas with ‎low levels of resistance cannot be generalized for other areas with high levels of ‎resistance (lack of external validity). This was expected to be included in the ‎discussion.”

Thank you for your sharp opinion. We agree, and we have added the following text to our discussion section.

Line 234

Furthermore, because responses to antibiotic treatment may be influenced by resistance patterns in specific geographical areas, the results of antibiotics treatment studies may not be generalizable across populations with vastly different levels of resistance. Our systematic literature was conducted without limiting for geographical area, clinical setting, local antibiotic resistance pattern, or language. Future studies of EATF should adjust the definition of symptoms according to the research target and attempt to account for the influence of local antibiotic resistance patterns.

“Second, due to the improvement of rapid diagnostic methods in ‎bacterial infections, such as the use of rapid detection methods for multi-‎resistant bacteria in blood cultures, these definitions can be different based on ‎Gram-negative (especially multi-drug resistant Gram-negative) and Gram-‎positive organisms.”

Thank you for your valuable suggestion.

Following your advice, we attempted to assess whether there were any distinctions made between Gram-negative bacteria (especially multidrug-resistant Gram-negative bacteria) and Gram-positive bacteria among the eligible studies. (ID 11,16,19,22,25,33,41,55,56. Tumbarello et al.(2010), Waltner-Toews et al. (2011), O’Neal et al.(2012), Kang et al. (2013), Hsieh et al.(2016), Lee et al.(2017), Herrmann et al. (2021), Mun et al.(2021)). We focused on bacteremia because both Gram-positive and Gram-negative bacteria are approximately equally likely to be the causative pathogen. However, we did not find any such distinctions in the papers included in this study. We therefore suggest the possible utility of such distinctions in future research.

According to your advice, we have added the following sentences in our discussion section.

Line 256

First, improved rapid diagnostic methods now enable clinicians to quickly find multidrug-resistant organisms in blood cultures. Among the eligible studies discussing bacteremia, there were no differences in definitions for EATF based on whether the involved organism is Gram-positive or Gram-negative. However, responses to antibiotics and clinical courses are often different for Gram-positive and Gram-negative bacteria. Thus, developing different definitions for EATF based on Gram-negative bacteria (especially multidrug-resistant strains) and Gram-positive bacteria may be clinically useful.

“Third, bacterial infections are also divided based on the ‎bacterial inoculum size and therefore should be considered in the early ‎antibiotic’s treatment failure definition. While urinary tract infections have a ‎low bacterial inoculum size, pneumonia and sepsis have a high bacterial ‎inoculum size, and thus the response to treatment may be different. Overall, for ‎serious bacterial infections more stringent definitions should be considered; ‎however, for milder infections, more flexible definitions may be acceptable. ‎There is no discussion in this regard in the manuscript and it is recommended ‎o improve the discussion section accordingly. Also, if possible, a table can be ‎prepared (based on the definitions of bacterial inoculum size) for more precise ‎comparison.”

Thank you for your sharp comment.

In order to evaluate the significance of bacterial inoculum in our study, we once again reviewed the EATF definitions in the 61 included studies. However, no study had a definition that directly referred to bacterial inoculum or bacteria count as a factor for EATF.

However, Mun et al. (2021) (ID 56) did suggest the clinical importance of high bacteria count:

"Furthermore, the presence of a high burden may also contribute to an inoculum effect, which attenuates the activity of beta-lactam antibiotics and often requires source control"

Based on your comment, we have added the discussion of this topic below.

Line 262

Second, while bacterial count may influence clinical outcomes in some cases, none of the 61 eligible studies included bacterial count as a factor for EATF. However, Mun et al. (2021) suggests the possibility of bacterial count affecting clinical outcomes and the importance of source control with some organisms. [16] Future studies may evaluate the importance of bacterial count and its potential relevance for assessing EATF.

“Although the findings of this study could be helpful there are ‎many shortcomings in the definition, which will make it difficult to reach an ‎accurate definition based on the wide variety of bacterial infections‏.‏”

We appreciate your point – the variety of bacterial infections is an important limitation to the usefulness of a standard definition for EATF. We have added this to our discussion.

Line 267

Lastly, this study comprehensively collected a diverse set of definitions for EATF from various studies. While each of these has its advantages, such as often being tailored to the target disease, they also have distinct disadvantages, such as being difficult to define based on observation of symptoms or other objective measures, and being difficult to apply to diseases other than the target disease. Therefore, a precise universal definition that can be applied to all types of bacterial infections may be difficult to achieve. Accordingly, the present study attempts to identify the essential components of a definition for EATF, rather than prescribing a definitive formulation.

Reviewer Comments

Reviewer #1

“Congratulations on the paper. The subject is exciting, and a lot of work to make this review is apparent.”

Thank you for your kind words. We appreciate your review of our work.

“- Elaborate further about the potential uses of EATF. However, this is not the purpose of the review; discussing the possible uses would be helpful for readers to integrate the concept into their work. Also, clarifying the potential uses even has implications for the components of the definition. For example, the use of EATF as an outcome for RCTs for new antibiotics has essential differences when compared with the use for clinical or antimicrobial stewardship purposes. For the second, component mortality would be of little help.”

Thank you for your sharp comment.

In addition to discussing potential clinical applications for EATF in the second paragraph of the discussion section, we have added the following sentence. Furthermore, your suggestion reminds us that the appropriateness of the various EATF components depends on the application. For example, the death component would be useless when informing treatment decisions in clinical practice.

Line 212

EATF may also have applications in the assessment of outcomes in antimicrobial drug trials and the establishment of criteria for early discontinuation. Furthermore, the appropriateness of the EATF components depends on the application. For example, the death component would be of no value in informing clinical treatment decisions.

“- From an antimicrobial stewardship point of view, using EATF raises some concerns that should be addressed. For example, analyzing together studies that report infections that need source control and those that typically do not need may create a concept (EATF) that promotes antibiotic escalation when the problem is source control. The authors should separate their findings for infections that typically do not need source control, such as respiratory tract infections and those that usually need source control.”

We appreciate this important consideration. 

While we agree that this is an issue which must be considered, we do not believe that this issue affected the contents of our proposed definition for EATF. First, we excluded "studies whose target disease’s standard treatment is not antibiotics," such as necrotizing enteritis requiring immediate operation. Also, we checked for " Source control (Drainage or operation) " in Table 2. Definitions for diseases that may need additional source control treatment may differ from those for diseases that typically do not. We counted the diseases which need souse control again from the table 1.We identified eight studies handling infections which may need additional source control treatment and marked them in Table 1. We considered analyzing two separate groups – studies of infections that need source control and those that typically do not. However, the eight studies, which included complicated skin and skin-structure infections, complicated urinary tract infection, acute appendicitis, intra-abdominal infection, and diverticular abscess, represent various diseases and definitions. Among the eight studies, only Edelsberg et al. (2008) and Berger et al. (2013) mention EATF at 72 hours, and thus were already included in our final assessment (see Table 2); however, the definitions of EATF used by Edelsberg et al. and Berger et al. (2013) did not influence the essential EATF components which we present. 

We have added the following discussions of the potential impact of source control on the application of EATF.

Line 198

Eight studies reporting infections which may need additional source control were identified, including complicated skin and skin-structure infection, complicated urinary tract infection, acute appendicitis, intra-abdominal infection, and diverticular abscess. However, because these infections can be successfully treated with antibiotics, these eight studies were not excluded from our analysis.

Line 242

Finally, because this study excluded target diseases whose standard treatment is not antibiotics, source control was not included among the factors for analyzing EATF which we identified. If the concept of EATF were expanded to cover all types of bacterial infections, source control would be an indispensable consideration from the perspective of avoiding antibiotic escalation.

“- Elaborate further about future steps to develop this interesting EATF concept and how it can be studied and used in future. Would the authors suggest some consensus procedure to define the concept? Delphi, for example?”

We agree with your suggestion about seeking consensus using the Delphi method. 

We added the description below, and also a new reference (Yoshida M. 2018).

Line 219

Furthermore, as the concept of EATF continues to be refined, methods for reaching consensus on a definition should be also considered. One possibility is the Delphi method, which facilitates consensus-building and minimizes the influences of potential sources of bias such as conflicts of interest and interpersonal relationships.

Reference

Yoshida M. Formulating Consensus for the Development of Clinical Practice Guidelines using the Delphi Method. J Tokyo Wom Med Univ. 2018;88: E35–E37.

“Minor revision:

- Page 21, line 223: mortality is misspelt.”

Thank you for finding this spelling error. It has been corrected.

Reviewer #2

“Italicize "et al." and names of bacteria

Prefer passive forms.

Numbers less than or equal to 12 should be spelled out if appropriate.”

Thank you for these directions. We have italicized “et al.” and names of bacteria, used passive forms where practical, and spelled out numbers less than or equal to 12 as appropriate.

“Why didn't the authors use EMBASE?”

We agree that EMBASE is a useful resource which may have allowed us to uncover additional candidate articles. We have added this as a limitation of our study. (See below.) Because this work received no specific funding, we were limited in terms of the resources at our disposal. 

In order to overcome our lack of access to EMBASE, we employed a range of search tools, including manual searches of Google and Google Scholar which were conducted by two independent teams with Dr. Aoki (our third author) overseeing the process. In addition, we sought help from an experienced librarian who helped us to craft search terms and formulas in order to minimize the chances of missing a potentially eligible article. We also conducted manual searches for papers referenced in reviews and other articles mentioning EATF. 

Bramer et al. (2017) surveyed combinations of search tools and reported that "[t]he highest scoring database combination without Embase is a combination of MEDLINE, Web of Science, and Google Scholar." We included all of these tools.

As a result, we uncovered approximately 2,500 candidate papers. While it is certainly possible that we missed some potential candidates, for example, because some texts might discuss concepts equivalent to EATF without using any of our search terms, we believe that we uncovered a broad enough range of papers discussing this topic to allow us to examine the conceptual framework of EATF.

Line 280

Next, the literature search was not conducted using EMBASE, due to financial constraints. In order to overcome this limitation, we employed a range of common systematic literature search methods. An experienced research librarian provided assistance with the selection of search terms and formulae, and two independent teams conducted manual searches on Google and Google Scholar. We also searched for all papers referenced in the reviews and articles that were uncovered with mentions of EATF.

Reference

Bramer WM, Rethlefsen ML, Kleijnen J, Franco OH. Optimal database combinations for literature searches in systematic reviews: a prospective exploratory study. Syst Rev. 2017;6: 245.

The authors would like to thank you again for taking the time to review our manuscript. We sincerely hope that you will consider this revised draft favorably and find it acceptable for publication.

Cordially yours,

Hiroyoshi Iwata, MD, MSc, PhD, FACP

Center for Environmental and Health Sciences, Hokkaido University

Kitaku Kita 12 Nishi 7, Sapporo, 060-0812, Hokkaido, Japan

Telephone: +81-11-706-4747 

E-mail1: hiwata@cehs.hokudai.ac.jp

E-mail2: hii887@mail.harvard.edu

---

## [Editor Report · Decision Letter 1]

6 Mar 2023

PONE-D-22-31074R1Essential components of a definition for early antibiotic treatment failure: a scoping review.PLOS ONE

Dear Dr. Hiroyoshi Iwata,

Thank you for submitting your manuscript to PLOS ONE. After careful consideration, we feel that it has merit but does not fully meet PLOS ONE’s publication criteria as it currently stands. Therefore, we invite you to submit a revised version of the manuscript that addresses the points raised during the review process.

We look forward to receiving your revised manuscript.

Kind regards,

Ali Amanati

Academic Editor

PLOS ONE

Additional Editor Comments:

Dear authors

The manuscript's overall presentation improved after amendments and is now ‎more readable‎. I thank the authors for their very detailed ‎replies to my comments.‎

Minor correction is needed:

Line 267: add "(inoculum effect)" after "bacterial count". "Future studies may evaluate the importance of bacterial count (inoculum effect) and its ..."

---

## [Author Response · Author response to Decision Letter 1]

6 Mar 2023

Dr. Ali Amanati

March 6, 2023

Dear Dr. Amanati,

On behalf of all of the authors, I would like to sincerely thank you and your team for your comments and suggestions for the improvement of our manuscript. We have incorporated all of your suggestions, as detailed below.

Comments

#1

The manuscript's overall presentation improved after amendments and is now ‎more readable‎. I thank the authors for their very detailed ‎replies to my comments.‎

Thank you for your support throughout this process.

Minor correction is needed:

Line 267: add "(inoculum effect)" after "bacterial count". "Future studies may evaluate the importance of bacterial count (inoculum effect) and its ..."

Response:

Thank you for your suggestion. We agree, and we have added "(inoculum effect)" after "bacterial count" in the manuscript.

Line 267

Future studies may evaluate the importance of bacterial count (inoculum effect) and its potential relevance for assessing EATF.

#2

While revising your submission, please upload your figure files to the Preflight Analysis and Conversion Engine (PACE) digital diagnostic tool, https://pacev2.apexcovantage.com/.

Response:

We appreciate your kind support. Following your instructions, we have made adjustments to "Figure 1" using the PACE tool, and re-uploaded it on the submission page.

The authors would like to thank you again for taking the time to review our manuscript. We sincerely hope that you will consider this revised draft favorably and find it acceptable for publication.

Cordially yours,

Hiroyoshi Iwata, MD, MSc, PhD, FACP

Center for Environmental and Health Sciences, Hokkaido University

Kitaku Kita 12 Nishi 7, Sapporo, 060-0812, Hokkaido, Japan

Telephone: +81-11-706-4747 

E-mail1: hiwata@cehs.hokudai.ac.jp

E-mail2: hii887@mail.harvard.edu

---

## [Editor Report · Decision Letter 2]

8 Mar 2023

Essential components of a definition for early antibiotic treatment failure: a scoping review.

PONE-D-22-31074R2

Dear Hiroyoshi Iwata,

We’re pleased to inform you that your manuscript has been judged scientifically suitable for publication and will be formally accepted for publication once it meets all outstanding technical requirements.

Kind regards,

Ali Amanati

Academic Editor

PLOS ONE

Additional Editor Comments (optional):

I read the revised manuscript ‎

I have no further comments to add. I thank the authors for their very detailed ‎‎replies to my comments.‎
---

## [Editor Report · Acceptance letter]

15 Mar 2023

PONE-D-22-31074R2 

Essential components of a definition for early antibiotic treatment failure: a scoping review. 

Dear Dr. Iwata:

I'm pleased to inform you that your manuscript has been deemed suitable for publication in PLOS ONE. Congratulations! Your manuscript is now with our production department. 

Kind regards, 

on behalf of

Professor Ali Amanati 

Academic Editor

PLOS ONE